# Outcomes in Patients Admitted for Upper Gastrointestinal Bleeding and COVID-19 Infection: A Study of Two Years of the Pandemic

**DOI:** 10.3390/life13040890

**Published:** 2023-03-27

**Authors:** Sergiu Marian Cazacu, Daniela Elena Burtea, Vlad Florin Iovănescu, Dan Nicolae Florescu, Sevastița Iordache, Adina Turcu-Stiolica, Victor Mihai Sacerdotianu, Bogdan Silviu Ungureanu

**Affiliations:** 1Research Center of Gastroenterology and Hepatology, University of Medicine and Pharmacy Craiova, 200349 Craiova, Romania; 2Department of Pharmacoeconomics, University of Medicine and Pharmacy of Craiova, 200349 Craiova, Romania

**Keywords:** COVID-19, upper gastrointestinal bleeding, ulcer, esophageal varices, endoscopy

## Abstract

Upper gastrointestinal bleeding (UGIB) represents a major emergency, and patient management requires endoscopic assessment to ensure appropriate treatment. The impact of COVID-19 on patient mortality in UGIB may be related to the combination of respiratory failure and severe bleeding and indirectly to delayed admissions or a reduction in endoscopic procedures. Methods: We conducted a retrospective study involving patients admitted between March 2020 and December 2021 with UGIB and confirmed. Our objective was to compare these types of patients with those negative for SARS-CoV-2 infection, as well as with a pre-pandemic group of patients admitted between May 2018 and December 2019. Results: Thirty-nine patients (4.7%) with UGIB had an active COVID-19 infection. A higher mortality rate (58.97%) and a high risk of death (OR 9.04, *p* < 0.0001) were noted in the COVID-19 pandemic, mostly because of respiratory failure; endoscopy was not performed in half of the cases. Admissions for UGIB have decreased by 23.7% during the pandemic. Conclusions: COVID-19 infection in patients admitted for UGIB was associated with a higher mortality rate because of respiratory failure and possible delays in or contraindications of treatment.

## 1. Introduction

The COVID-19 global pandemic appeared in China during the last days of 2019 and spread very fast in all countries during the spring of 2020 [1,2,3,4]. The pandemic has had multiple negative consequences on the healthcare system, which mainly focused on lockdown measures, the need for dedicated beds, additional personnel, specific access routes for SARS-CoV-2 patients, and overwhelmed emergency rooms [2,3]. Moreover, unclear protocols at the onset of the pandemic, secondary drug effects, and complications associated with prolonged stay in the ICU represent another part of the complicated management of these patients.

Acute upper gastrointestinal bleeding (UGIB) represents one of the most common emergencies, with an estimated mortality of 5–10% for non-variceal bleeding in most countries [5,6,7,8,9,10], although values as high as 14–15% have been estimated in other studies [11,12,13]. The mortality is usually correlated with bleeding severity and associated comorbidities [12,13,14], whereas in variceal bleeding, the liver disease stage seems to be the most relevant factor.

Endoscopic hemostasis remains the main therapeutic option [15] and should be performed as soon as possible after assessing the hemodynamic status. While most guidelines recommend an early (first 24 h) endoscopy for most patients with UGIB [16,17], when variceal bleeding is suspected, the interval should be smaller [18,19,20]. Upper digestive endoscopy is considered a high-risk procedure with potential aerosol dissemination, which made all practicians reluctant at first to perform the procedure without proper protection with sanitary materials [1,2]. Thus, risk management precautions should be instated to impact the timing and efficiency of endoscopic procedures. On the other hand, during the first phase of the pandemic, most patients avoided hospital presentations because of the “fear effect” of contamination risks [2], so a marked reduction in emergency department presentations and hospital admissions was noted [2,3,4,21,22,23]. Moreover, some hospitals instated their own protocols, which required COVID-19 testing before performing the procedures or focused on patient isolation. Thus, it negatively influenced the timing of endoscopy and potentially increased the mortality rate.

The association between UGIB and COVID-19 infection may represent a major diagnostic and therapeutic challenge. Severe COVID-19 infections are frequently associated with extensive thrombosis, which requires anticoagulant therapy. It is conceivable that the COVID-19 virus may cause direct damage to the mucosa that leads to an immune response, as well as indirect damage due to the hypoxic stress secondary to coagulopathy. Nonetheless, the use of non-steroidal anti-inflammatory drugs during this period may also increase the risk of potential bleeding [8,10].

Our study objective was to assess the mortality rate in patients admitted for UGIB who also had a COVID-19 infection. We assessed the pandemic’s effect on UGIB admissions, as well as the factors that influenced patients’ evolution in almost two years of the pandemic, in our endoscopic center.

## 2. Materials and Methods

We conducted a retrospective, single-center study of all patients admitted for UGIB within the Gastroenterology Department of the Emergency County Hospital of Craiova, Romania during 22 months of the pandemic.

### 2.1. Study Population

All patients presenting with non-variceal or variceal bleeding between March 2020 and December 2021 were compared to a control group before the COVID-19 period from March 2018 to December 2019. All cases were identified using the medical charts and the International Statistical Classification of Diseases and Related Health Problems (ICD), which allowed for patient selection with various diagnoses of UGIB, as well as COVID-19 infection. We also checked for patients admitted for other medical conditions who presented with a bleeding episode during their admission. We selected patients admitted for UGIB coded by ICD-10-AM as esophageal disease with bleeding (K22.0 and K22.2), gastric, duodenal, peptic, or gastro-jejunal ulcer with bleeding (K25.0, K25.4, K26.2, K27.2, K27.4, K28.0, K28.4), hemorrhagic gastritis or duodenitis (K29.0), bleeding angiodysplasia (K31.82), hematemesis (K92.0), melena (K92.1), gastrointestinal bleeding (K92.2), and esophageal varices with bleeding (I85.0); COVID-19 infections were selected as U.07.1.

### 2.2. Patient Characteristics

Informed consent was obtained from all patients for the endoscopic procedures, whereas the study was approved by the local ethical committee (11977/24 March 2020). All clinical, laboratory, and endoscopic information from patients with data were noted, and the analysis was made by using stratification risk factors for mortality, such as pre-endoscopic and post-endoscopic risk scores (Rockall, Glasgow-Blatchford, Baylor, Cedars-Sinai, AIM65, T-score) [6,11,12,13,14] for non-variceal bleeding and the Child-Pugh classification for variceal bleeding. We also assessed the interval between the onset of bleeding and admission and between admission and endoscopy. The Charlson comorbidity index was included as well in the analysis.

A protocol for patients admitted during the pandemic was applied in order to analyze the impact of the pandemic on admissions, general mortality, and the risk of death in patients with UGIB and active COVID-19 infection.

### 2.3. COVID-19 Assessment

During the first period (before the introduction of rapid antigen testing for COVID-19), the hospital-admitted patients were evaluated via epidemiologic triage (including symptom and data contact triage), thoracic X-ray, and PCR testing in suspected cases; most patients were examined via endoscopy in the first 24 h. Because of the lack of extensive PCR testing at the beginning of the pandemic, a protocol for endoscopy was implemented in all cases of UGIB, using protective FFP2 or FFP3 masks, 3 pairs of gloves, and surgical impermeable gowns. After the introduction of rapid antigen testing, all cases admitted to our hospital for gastrointestinal bleeding were assessed based on clinical symptoms, rapid antigen testing, and thoracic X-rays, and all suspected cases were tested by PCR and managed conservatively, if possible; hemodynamically unstable cases or with ongoing bleeding who were positive or suspected for COVID infection were evaluated via emergency endoscopy in a dedicated endoscopy room. Cases of COVID-19 infection admitted to other hospitals who subsequently developed UGIB were transferred to our hospital in case of significant bleeding.

### 2.4. Outcomes

The assessed outcomes were as follows: the rate of in-hospital mortality, the rate of re-bleeding and hemostasis failure, the need for transfusion and for surgery, and the hospitalization duration. The monthly rate of admission was compared before and during the pandemic. The analyzed risk factors were the proportion of variceal bleeding and cirrhotic patients, the severity of bleeding (prognostic scores for non-variceal bleeding and Child-Pugh score for variceal bleeding), the time interval between admission and endoscopy, and the onset of symptoms.

### 2.5. Statistical Analysis

Statistical data were analyzed and provided using GraphPad Prism 9.4.1 (GraphPad Software, San Diego, CA, USA). Continuous variables were compared using the Mann–Whitney test if they were found to have no normal (Gaussian) distribution after a test for distribution with the Kolmogorov–Smirnov normality test, while for categorical variables, a Chi-square test or Fisher test was used. We calculated the odds ratio (OR) to assess the strength of the association between the two events. Statistical significance was considered a *p*-value less than 0.05 with a two-tailed analysis.

## 3. Results

We analyzed data from 1905 patients; 1081 were admitted pre-pandemic and 824 were admitted during the COVID-19 pandemic (498 were tested).

### 3.1. SARS-CoV-2-Positive Versus Negative Patients

During the 22 months, 44 patients from 498 tested patients with UGIB tested positive for COVID-19 infection (a positivity rate of 8.8% for tested patients and 5.3% from all cases, slightly superior to the positivity rate of 5.67% in the general population of Romania and also to the positivity rate in Dolj county = 4.3%, *p*-value = NS). Thirty-nine cases were active COVID-19 infections (8% of tested cases and 4.73% of total UGIB cases) and five patients were post-COVID. Endoscopy was performed on only 17 patients with active COVID infection, and the other 22 patients were managed conservatively because of severe respiratory failure (56.4%). Therapeutic endoscopy was necessary for only four cases with two esophageal band ligations and two combined treatments for bleeding ulcers (23.5%). The average age was 67.1 years for patients with COVID-19 active infection and UGIB versus 63.1 for those with UGIB without COVID-19 (*p*-value = 0.0988). In addition, no difference was observed in age between the deceased patients with COVID-19 infection (mean = 70.1 years) and survivors (mean = 62.8) (*p*-value = 0.0817). Most patients with COVID-19 infection had significant comorbidities. In the group of COVID-tested patients, the mortality rate was significantly higher (59%) as compared with that in patients negative for COVID (13.7%, *p*-value < 0.0001) (Table 1).

When endoscopy was performed for SARS-CoV-2-positive cases, the bleeding source was dominated by ulcers (52.9%), followed by variceal bleeding and tumors (11.8% each). In 56.4% of positive cases, no endoscopy was performed, mainly because of respiratory failure, and all patients with no endoscopy performed showed no signs of hemodynamic instability (Table 2).

All patients with UGIB and COVID-19 infection were treated according to the available guidelines at the moment of diagnosis; remdesivir was not used in cases of cirrhosis. Anticoagulant therapy (low-weight molecular heparin) was used in severe and potentially severe cases after endoscopic or conservative hemostasis was achieved, and no rebleeding was noted.

During the analyzed pandemic period, 22 cases admitted for UGIB were in the potential vaccination period (January 2021–December 2022): 13 patients were not vaccinated, one patient was vaccinated, and one patient had COVID-19 infection 3 months before admission.

In COVID-positive cases, the Rockall pre-endoscopy, Glasgow-Blatchford score, and Charlson comorbidity index were higher than those in negative patients, which suggested more severe associated diseases in COVID-19 patients (many patients having COVID-19 pneumonia) (Table 3).

The presence of COVID-19 infection was associated with a significant risk of death (OR = 9.04, 95% CI 4.5265 to 18.0369, *p*-value < 0.0001). Only 2 of the 23 deaths were related to severe anemia, and most patients (14 patients of 23, meaning 61%) died because of respiratory failure. The mortality rate for COVID-19 patients was 58.97%. The Glasgow-Blatchford and Charlson comorbidity index was higher on average for COVID-19-positive patients who died compared to that in those who survived, while for Rockall pre-endoscopy and post-endoscopy, the differences were not statistically significant (Table 4).

### 3.2. UGIB during the Pandemic and Pre-Pandemic Period

Here, 824 patients with UGIB were included during the pandemic period and 1081 during the March 2018–December 2019 period, with an average monthly admission for UGIB of 49.1 during the pre-pandemic period and 37.5 for the pandemic period (23.7% reduction). The etiology of bleeding was similar (with no statistical difference). Although the global mortality rate was slightly higher during the pandemic, the difference was not statistically significant (13.71% versus 12.68% before the pandemic, *p*-value = 0.51, 95%CI 0.84–1.43). More patients had no endoscopy performed during the pandemic, and a higher proportion of patients needed blood transfusions. Both the time from onset to admission and from admission to endoscopy was higher (mean difference 7.3 h and 3.1 h, respectively) (Table 5 and Table 6).

We postulated that in periods with a higher number of new COVID-19 cases, more patients with UGIB (especially mild cases) were reluctant to go to the hospital, and therefore, we considered that there was a monthly variation in the difference between cases admitted during the pre-pandemic period compared with those admitted during the pandemic period, and we compared to the monthly variation of new cases in Romania during March 2020–December 2021 [24,25] (Figure 1). Although there was no strict alignment between the two graphs, we noted the first peak difference during March–May 2020 (lockdown) similar to that in other countries [2], and the other three peaks during February, May, and October 2021.

### 3.3. UGIB in Initially Non-Bleeding Hospitalized COVID-19 Patients

We analyzed the rate of UGIB for patients admitted for other pathologies and COVID-19 infection in our hospital. During the pandemic, our hospital received mainly COVID-positive patients admitted for other pathologies who tested positive at presentation in the emergency unit or were later confirmed with infection during hospitalization. Another group of patients came to the intensive care unit with a severe form of COVID infection, which required non-invasive or invasive ventilation. All of those patients were treated with low-molecular weight heparin (LWMH) and corticosteroids, which can increase the risk of gastrointestinal bleeding, together with hypoxemia in severe cases. Of 1881 patients admitted with COVID-19 infection, only 11 (0.58%) had UGIB, all treated with LWMH. The mortality rate was 54.5% in patients with UGIB and 45% in those without UGIB, with no statistically significant difference (*p*-value = 0.5271); 1025 patients stayed in the ICU for several days during admission (54.49%), many with severe respiratory failure and on mechanical ventilation.

## 4. Discussion

The COVID-19 pandemic had a major influence on the healthcare system, not only through the admissions caused by the disease but also the effects that were seen afterward with many patients postponing their medical examinations [3]. UGIB management and emergencies were a challenge for the physicians at the beginning of the pandemic until most endoscopy services were properly instated to ensure that precautions were taken for both patients and medical personnel.

Similar to other tertiary centers around the world, we had to ensure treatment for UGIB patients. We highlighted that in our study period, we had 39 patients with UGIB with active COVID-19 infection. Our results are similar to other studies published [4,26,27,28,29] so far, highlighting variceal bleeding or peptic ulcer as the most encountered. There was a general decrease noted for UGIB in some studies [4], especially for non-variceal bleeding, even though there is a predisposition for gastric ulceration and peptic ulcers in COVID-positive patients [30,31].

The presence of COVID-19 infection was associated with a significant risk of death (OR 9.04, 95% CI 4.5265 to 18.0369, *p*-value < 0.0001). A meta-analysis of eight studies showed a pooled mortality of 19.1% [27]. Nonetheless, COVID-19 patients with UGIB might have a higher death risk than patients with UGIB alone, due to the possible association with respiratory failure. Moreover, if a patient rebleeds, the mortality rate will increase. In our study, only one positive patient rebled, while in the literature, the rebleeding rate is 10–11.3% [27,29]. In a meta-analysis of four studies with 820 patients with COVID-19 and UGIB, an OR of 3.85 for death was estimated [32]. Although the risk for COVID-19 patients was much greater in our study, the relatively small number (4.7%) of infected patients produces only a limited effect on UGIB mortality.

Current guidelines for UGIB suggest that endoscopy might be postponed for the next 24 h if the patient is not hemodynamically unstable. Moreover, the initial assessment with pre-endoscopic scores is considered a valid option for these types of patients. However, some studies showed that Rockall and Glasgow-Blatchford scores were not correlated with mortality in COVID-positive UGIB [33]. The Rockall pre-endoscopy, Glasgow-Blatchford score, and Charlson comorbidity index were higher than those in positive patients, which perhaps was related to the associated pulmonary disease, as most of them developed COVID-19 pneumonia [33].

To determine if there were more severe cases during the pandemic, we analyzed the proportion of patients with variceal bleeding and liver cirrhosis and the average prognostic score using all scores used for stratification. The percentage of variceal bleeding and percent of patients with cirrhosis were similar between the two groups of patients. However, some scores (Glasgow-Blatchford, Rockall pre-endoscopic and after endoscopy, Baylor pre-endoscopic and after endoscopy) had a slightly higher mean value during the pandemic. These findings may suggest a possible superior proportion of severe cases during the pandemic compared to that in pre-pandemic cases. The analysis of the Charlson comorbidities index (CCI) score showed, however, that mean values for patients with UGIB during the pandemic and before the pandemic were similar (3.88 versus 3.76, *p*-value = 0.2194).

The impact of the COVID-19 pandemic on endoscopies was felt worldwide, with a global reduction in endoscopic procedures of 50% in a study that included 48 countries [21]. Similarly, a reduction in endoscopies for UGIB was noted in China, with 73.4% during their lockdown period [2] and 70.3% in a Romanian center for 9 months [3]. In our study, a 23.7% reduction in UGIB admission was noted; reluctance for patients to come to the hospital may be the main explanation for that reduction in admissions. This is similar to that in other countries; in Austria, a 40% reduction in hospitalization for UGIB has been observed during the lockdown [4] and in Hubei, China, a 55.7% reduction in UGIB was noted [2]. In a study in Hong Kong, where there was no lockdown, there was an inverse correlation between the number of cases and hospitalizations for UGIB [34], similar to that in our study.

Most endoscopy centers preferred a more pragmatic approach [1,22,26,27], choosing to perform the procedure in COVID-positive cases within 24 h of admission only for patients with hemodynamic instability and with the inability to maintain their Hb level [1,15,26]. On the other hand, respiratory failure was the main contraindication [15], and some studies advocated, as an indication, a Glasgow-Blatchford of score ≥ 7 [28]. Mauro et al. [28] reported that only 48% of endoscopies were performed in the first 24 h and endoscopic treatment was necessary in 39–40% of the cases. Others stated that the rate of endoscopy in COVID-positive cases ranged from 32 to 78.26% [1,6,29], and a conservative approach was recommended for some COVID-positive patients [1,26,27], because the 30-day mortality was not altered by the delay of endoscopy for 24 h [15,30].

Similar global mortality, hospital stays, rebleeding rates, and failure of endoscopic hemostasis were noted in our study in pandemic and pre-pandemic periods; however, a higher mean onset time and time-to-endoscopy were observed during the pandemic period. Medical personnel and patients were clearly affected by the COVID-19 pandemic; however, when discussing UGIB mortality, the status is still unclear. Although patients with SARS-CoV-2 infection and comorbidities may have an increased mortality rate, some centers showed similar pandemic and pre-pandemic results with a similar rate of rebleeding and mortality [24,25]. When discussing patients admitted for COVID-19 infection or for other diseases (with the exception of UGIB), the rate of bleeding during admission was very low (0.58%), which is at the lower end of the literature data, 0.4–13% [1,28].

We acknowledge that our study has several limitations. First of all, this is a retrospective, single-center study, and a small proportion of our cases have not been completely investigated in order to gather all information for clinical and endoscopic scores. Endoscopy was not performed in 18.2% of the cases, so post-endoscopic scores were not available. A significant proportion of patients was not tested for COVID-19 during the early phases of the pandemic because of the lack of PCR-testing facilities. An autopsy was not performed on deceased patients with COVID-19 infection, according to the national protocol at that time. Moreover, we did not have information on the COVID-19 severity according to the WHO classification (WHO/2019-nCOV/clinical2020.5) by level of disease severity: mild, moderate, severe, and critical.

## 5. Conclusions

Our retrospective study emphasizes the need for individualized therapy during the pandemic when resources were limited. We highlighted that positive COVID-19 patients with UGIB had a much higher risk of death and the impact of the pandemic on both patients and the healthcare system was significant especially with the first positive COVID-19 cases. A higher time from onset to admission and also from admission to endoscopy was noted, which can suggest a delay in presentation for some cases. Further large studies are required to prevent similar situations, and perhaps multicenter studies would aid in covering the flaws encountered at the start of the COVID-19 pandemic.

## Figures and Tables

**Figure 1 life-13-00890-f001:**
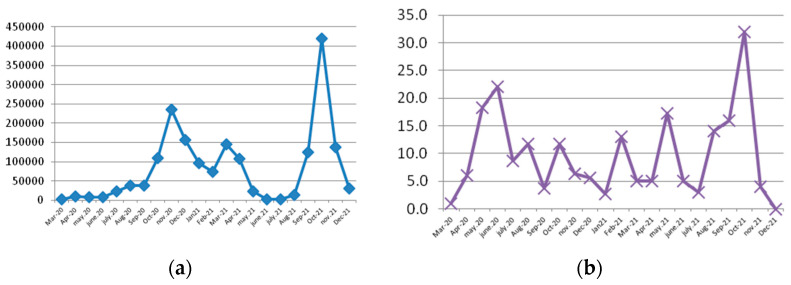
(**a**) Difference in mean monthly admissions; (**b**) new COVID cases in Romania, March 2020–December 2021.

**Table 1 life-13-00890-t001:** Main outcomes of UGIB in SARS-CoV-2-positive versus negative patients.

	Positive No = 39	Negative No = 459	Odds Ratio	95% CI	*p*-Value
Mortality%	58.97	13.73	9.04	4.53 to 18.04	<0.0001
Rebleeding%	2.56	7.84	0.31	0.04 to 2.32	0.2535
Surgery%	5.13	2.18	2.43	0.51 to 11.49	0.2637
No of blood units transfused	3.4	3.3	-	−0.73 to 0.81	0.9169
Hospital admission days	10.1	6.9	-	0.32 to 6.02	0.0026

Abbreviations: CI—confidence interval.

**Table 2 life-13-00890-t002:** UGIB SARS-CoV-2-positive and negative patient characteristics during the pandemic period.

	SARS-CoV-2 Positive No = 39	SARS-CoV-2 Negative No = 459	*p*-Value
Age (yrs)	67.8 (41–86)	63 (16–99)	0.0988
<60/60–79/>80 (%)	23.1/56.4/20.5	36.8/48.6/14.6	0.2037
M/F (%)	56.4/43.6	68/32	0.1433
Etiology (%)			0.1675
Ulcer	47	43.9	
Erosions	0	5.5	
Mallory-Weiss	0	12.8	
Varices	11.8	19.8	
Tumor	11.8	7	
Antithrombotic	11.8	3.7	
Other	17.6	7.3	
Mortality	59	13.7	<0.0001
Endoscopy Yes/No (%)	17 (43.6)	384 (83.7)	<0.0001
Endoscopic treatment (%)	4 (23.5%)	92 (24)	0.9677
Variceal (No of procedures)	EVL (2)	EVL (44)	
Non-variceal	Adrenaline + other (2)	Adrenaline + other (48)	
Comorbidities (%)			
Cardiac	12.8	17	0.5038
Renal	10.3	5.9	0.5363
Cirrhosis	38.5	29.4	0.2395

Abbreviations: EVL—esophageal variceal ligation.

**Table 3 life-13-00890-t003:** Mean prognostic scores in COVID-positive and negative patients.

Score	SARS-CoV-2 Positive (SD, No) *n* = 39	SARS-CoV-2 Negative (SD, No) *n* = 459	*p*-Value
Rockall pre-endoscopy	3.77 (1.40, 35)	3.07 (1.70, 430)	**0.0189**
Rockall post-endoscopy	4.38 (2.22, 13)	4.48 (2.04, 360)	0.8641
Glasgow-Blatchford	12.2 (2.81, 35)	10.81 (3.65, 427)	**0.0278**
Charlson comorbidity index	4.85 (2.21, 39)	3.97 (2.26, 459)	**0.0203**

Abbreviations: SD—standard deviation. Significant values are in bold. A Mann–Whitney test was used.

**Table 4 life-13-00890-t004:** Mean prognostic scores in COVID-positive patients who died versus those who survived.

Score Mean ± SD	Deceased (SD, No) *n* = 23	Survivors (SD, No) *n* = 16	*p*-Value
Rockall pre-endoscopy	4.05 (1.40, 21)	3.36 (1.34, 14)	0.1543
Rockall post-endoscopy	5.00 (2.61, 6)	3.86 (1.86, 6)	0.3774
Glasgow-Blatchford	13.24 (2.14, 21)	10.64 (3.03, 14)	**0.0055**
Charlson comorbidity index	5.61 (2.06, 20)	3.75 (1.98, 14)	**0.0078**

Abbreviations: SD—standard deviation. Significant values are in bold. A Mann–Whitney test was used.

**Table 5 life-13-00890-t005:** UGIB patient characteristics during pandemic versus pre-pandemic periods.

	Pandemic (*n* = 824)	Pre-Pandemic (*n* = 1081)	*p*-Value
Age (years)	62.6 (20–95)	63 (16–99)	0.55
<60/60–79/>80 (%)	40/48.5/11.5	35.8/53.4/10.8	0.11
M/F (%)			
Hb (g/dl)	67.6/32.4	63.8/36.2	0.19
Ht (%)	8.5	8.68	0.16
Etiology (%)			
Ulcer	38.1	39.2	0.12
Erosions	12.7	13.5	0.30
Mallory-Weiss	6.7	7	0.53
Varices (E/G/J)	22.3 (168/16/0)	23.4 (234/18/1)	0.19
Tumor	7	6.1	0.74
Antithrombotic	4.9	3.7	0.18
Other	8.3	7.1	0.32
Mortality (%)	13.71	12.68	0.51
Endoscopy <6/<12/<24/NO	39.9/59.9/85.3/18.2	46.6/64.1/86.5/11.7	**0.0001**
Endoscopic treatment (%)	64% (V), 13.9% (NV)	58.3% (V), 16.4% (NV)	0.24/0.24
Blood transfusions %/mean	58/3.4	49.7/3.21	**0.0003**
Rebleeding rate (%)	5.1	3.9	0.20
Failure of hemostasis (%)	1.46	1.3	0.76
Emergency surgery (%)	1.33	0.65	0.13
Hospital admission days (mean ± SD)	7.01 ± 5.85	7.31 ± 6.14	0.26
Hours from onset to admission time (mean ± SD)	56.2 ± 55.2	48.9 ± 63.1	**0.006**
Hours from admission to endoscopy (mean ± SD)	18.9 ± 34.76	15.8 ± 28.33	**0.045**

Abbreviations: Hb—hemoglobin, Ht—hematocrite, E—esophageal, G—gastric, J—jejunal, NV—non variceal, V—variceal, SD—standard deviation. Significant values are in bold. The Mann–Whitney test was used for continuous variables and a Chi-square test was used for categorical variables.

**Table 6 life-13-00890-t006:** Evaluation of case severity during the pandemic compared to the previous period.

Severity Factors	Pandemic *n* = 824	Pre-Pandemic *n* = 1081	*p*-Value
Variceal bleeding (%)	22.3%	23.4%	0.19
Cirrhosis (%)	31.4%	32.1%	0.76
Scores (mean ± SD)			
Glasgow Blatchford	11.02 ± 3.61	10.63 ± 3.76	**0.0268**
Rockall pre-endoscopy	3 ± 1.63	3.25 ± 1.61	**0.0015**
Cedars Sinai	4.23 ± 2.45	4 ± 2.56	0.0920
Baylor pre-endoscopy	9.27 ± 3.65	8.13 ± 3.82	**<0.0001**
Baylor post-endoscopy	10.18 ± 4.15	9.165 ± 4.24	**<0.0001**
AIM65	1.38 ± 0.89	1.25 ± 1.06	0.1358
T-score	11.34 ± 2.12	11.18 ± 2.08	0.1212
Rockall post-endoscopy	4.46 ± 2.06	4.82 ± 2.02	**0.0008**

Significant values are in bold. A Mann–Whitney test was used for continuous variables and a Chi-square test was used for categorical variables.

## Data Availability

Not applicable.

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
