# Peer review of "Outcomes in Patients Admitted for Upper Gastrointestinal Bleeding and COVID-19 Infection: A Study of Two Years of the Pandemic"

_life, 2023, doi:10.3390/life13040890_

Round 1

Reviewer 1 Report

The authors aimed to analyze different aspects of upper gastrointestinal bleeding in a tertiary center during the COVID-19 pandemic. The data presented are interesting and seem more suitable for a journal dedicated to gastrointestinal diseases.

The title reflects only one issue from the results of this study: mortality. I think that the study would better aim to explain (with data and analysis) different factors that influence the increase of mortality in COVID-19 patients, especially as there is no significant difference in pre-pandemic and pandemic patients. There is any relationship between the lack of endoscopy or the time  from onset to admission and to endoscopy and the mortality? Or just the severity of disease in COVID-19 patients is the cause.... The title may be improved to cover all the issues discussed in this paper, not only mortality.

In Material and methods there is no description of the factors included in different scores. Moreover, there is no info regarding Charlson comorbidity index. Child-Pugh classification was not use anymore in presentation of the results. This section must be improved.

If you verify the results, lines 174-176 and Table 2 there is a contradiction regarding Charlson comorbidity index. It is lower in patients with COVID-19 who died, and hot higher as you interpret it.

In Discussion, there should be a repetition of the results, but only the analysis and explanation of them correlated with other similar studies. Please verify the lines 223-224 as the data are somehow different from those presented in results.

Some references are cited without important significance (see self-citations 33,34). There should be more consistent information coming from those papers to be included in the discussions. 

In Conclusions there should be no results presented again, just the significant conclusions based on the results of the study.

Author Response

We are very grateful to the constructive comments from you. We also thank you for the time and effort on reviewing our manuscript. We have carefully addressed point-by-point all the comments and made corrections in our manuscript using tracked changes.

The authors aimed to analyze different aspects of upper gastrointestinal bleeding in a tertiary center during the COVID-19 pandemic. The data presented are interesting and seem more suitable for a journal dedicated to gastrointestinal diseases.

The title reflects only one issue from the results of this study: mortality. I think that the study would better aim to explain (with data and analysis) different factors that influence the increase of mortality in COVID-19 patients, especially as there is no significant difference in pre-pandemic and pandemic patients. There is any relationship between the lack of endoscopy or the time  from onset to admission and to endoscopy and the mortality? Or just the severity of disease in COVID-19 patients is the cause....

We appreciate your question very much. We checked the possible relationship between the lack of endoscopy, the time from onset to admission and to endoscopy, the severity of disease and the mortality. The multivariate logistic regression model was used to identify independent risk factors for mortality and the adjusted odds ratio (AOR) was calculated. In our multivariate analysis, we did not find any variables independently associated with mortality of COVID-19 patients (AOR for endoscopy was -0.442, 95%CI 0.177-2.333, p-value=0.502; AOR for the time from onset to admission and to endoscopy was 0.092, 95%CI 0.964-1.246, p-value=0.162). We did not have information on the COVID-19 severity according to the WHO classification (WHO/2019-nCoV/clinical/2020.5) by level of disease severity: mild, moderate, severe, and critical [World Health Organization. Clinical Management of COVID-19. 2020. Available online: www.who.int/publications/i/item/clinical-management-of-covid-19].

The title may be improved to cover all the issues discussed in this paper, not only mortality.

In Material and methods there is no description of the factors included in different scores. Moreover, there is no info regarding Charlson comorbidity index. Child-Pugh classification was not use anymore in presentation of the results. This section must be improved.

We appreciate your comment. We modified and included the Charlson Comorbidity in the patient’s characteristics section. Whereas we did not describe the factors included in the endoscopic scores because they are guideline-included.

If you verify the results, lines 174-176 and Table 2 there is a contradiction regarding Charlson comorbidity index. It is lower in patients with COVID-19 who died, and hot higher as you interpret it.

We agree with you and correct the mistake. Thank you for your observation.

In Discussion, there should be a repetition of the results, but only the analysis and explanation of them correlated with other similar studies. Please verify the lines 223-224 as the data are somehow different from those presented in results.

We agree with your observation. We modified the entire section from the Discussion chapter as suggested.

Some references are cited without important significance (see self-citations 33,34). There should be more consistent information coming from those papers to be included in the discussions. 

We decided to remove the references as suggested.

In Conclusions there should be no results presented again, just the significant conclusions based on the results of the study.

Thank you for your observation, we modified the conclusion section.

Reviewer 2 Report

In my opinion, the paper was not satisfactorily clear, informative and currently does not provide a valuable source document for anyone requiring a primer to know and understand this issue. Namely, numerous shortcomings in the section Abstract, Materials and Methods, Results, Discussion and Conclusions make this paper not appropriate for publication in this form. Some comments:        Line 19: In this sentence it is stated `A prospective study ..`. But, in the section Materials and Methods in the text of this manuscript on Line 77 it is stated `We conducted a cross-sectional, observational, ...`. Further, on Line 310 (in the section Conclusions) it is stated `Our retrospective study ...`. What is actually true and correct, which Study design was actually applied in this manuscript?     Line 24: The mentioned data `high risk of death (OR 9.04, p <0.0001` was not presented in any of the Tables in this manuscript. Explain.      Line 24: In the text of the manuscript (nor in any of the Tables in this manuscript) the mentioned data `mortality rate (55.2%)` was not presented. Explain.     Line 24: Parameter `OR` is stated in the Abstract (i.e., as a result of this manuscript), but it is not mentioned in the subsection `Statistical analysis` and those methods that are mentioned as used in the statistical analysis do not allow for `OR` to be determined. `OR` shold most probably represent an abbreviation for `Odds Ratio`, but the statistical methodology for determination of `OR` was not applied in this manuscript. Further: the stated data for `OR` is given in the text of this manuscript, and in the section Discussion and in the section Conclusions, even though it was not presented in the section Results.      Lines 67-68: In the cited reference No. 8 there is no mention of `non-steroidal anti-inflammatory drugs` at all? Explain.      Lines 70-73: The manuscript does not present data on `... death causes in patients admitted for UGIB who also had a COVID-19 infection.`, therefore that aim was not achieved.     Lines 84-88: State all codes, according to ICD, which were used to identify the cases. State which version of the ICD was used.     Line 91: First of all, it should be explained in detail in the section Materials and Methods whether the patients provided WRITTEN VOLUNTARY consent to participate in the study, or whether they only gave an `Informed consent` for the endoscopic procedures.      Line 98: Add to the description of the characteristics of patients also the values of laboratory parameters (such as mean Hb level, etc), which are of importance to the study or are mentioned multiple times in the Discussion in this paper and are not presented in the Results.       Line 135: In the entire section Results, below every table and figure the applied statistical test should be stated.     Lines 138-163: The stated data are directly associated to the aims of this manuscript, therefore they had to be systematized and presented in detail on an appropriate Table.      Line 164-166: Provide data for the vaccinal status of the stated 22 patients.     Lines 174-178: On Table 2, alike the Table 1, make sure to mention the NUMBER OF PATIENTS for every score, i.e. `mean (SD, No)`.            Lines 174-178: Since `... to assess ... death causes` was stated as an aim of the paper (on Line 70), the causes of death must be stated in Table 2.      Lines 174-178: Results presented in Table 2 are not described appropriately in the corresponding text.         Lines 179-188, 198-200, 202-204: The causes of death are not stated.      Lines 189-197: In the subsection Statistical analysis not one test for parallelism was stated. State the results of the test of parallelism.      Lines 208-221: In the context of the above mentioned patients description, present the characteristics of patients to which this paragraph refers to on a Table.      Lines 222-308: Reconstruct the entire text of the section Discussion in a way that explanations for differences and similarities in the results of this manuscript and findings of other cited studies are provided.    Lines 230-231: If the data presented in this sentence represent a part of this manuscript, the statistical methodology described in the subsection Statistical analysis does not allow the use of the term `associated`. Explain.      Lines 231-232: In the section Results in this manuscript it is stated that 23 patients died (see Table 2). In this sentence `Only 2 of the 17 death cases ...` are described. Explain.        Lines 244-247: State in which study were the values recorded as mentioned in this sentence.      Lines 276: In the Discussion, explain why in this study the COVID-19 patients with a significantly lower mean score for Charlson comorbidity index died, and those patients with a significantly higher mean score for Charlson comorbidity index survived COVID-19.        Lines 312-314: The stated data are not presented in this manuscript. Explain.         

Author Response

Reviewer 2

Dear reviewer, we are very grateful for the constructive comments from you. We really appreciate your effort in reviewing our paper which clearly helped us to improve our manuscript.  We have carefully addressed point-by-point all the comments and made corrections in our manuscript using tracked changes.

In my opinion, the paper was not satisfactorily clear, informative and currently does not provide a valuable source document for anyone requiring a primer to know and understand this issue. Namely, numerous shortcomings in the section Abstract, Materials and Methods, Results, Discussion and Conclusions make this paper not appropriate for publication in this form. Some comments:       

Line 19: In this sentence it is stated `A prospective study ..`. But, in the section Materials and Methods in the text of this manuscript on Line 77 it is stated `We conducted a cross-sectional, observational, ...`. Further, on Line 310 (in the section Conclusions) it is stated `Our retrospective study ...`. What is actually true and correct, which Study design was actually applied in this manuscript?    

Thank you for your observation. The study is retrospective, and we modified our mistakes.

Line 24: The mentioned data `high risk of death (OR 9.04, p <0.0001` was not presented in any of the Tables in this manuscript. Explain.  

We included the OR into results and discussion sections. Thank you.  

Line 24: In the text of the manuscript (nor in any of the Tables in this manuscript) the mentioned data `mortality rate (55.2%)` was not presented. Explain.    

We agree the mortality rate was not presented in the Results section and we introduce it before the Table 2. Thank you.

Line 24: Parameter `OR` is stated in the Abstract (i.e., as a result of this manuscript), but it is not mentioned in the subsection `Statistical analysis` and those methods that are mentioned as used in the statistical analysis do not allow for `OR` to be determined. `OR` shold most probably represent an abbreviation for `Odds Ratio`, but the statistical methodology for determination of `OR` was not applied in this manuscript. Further: the stated data for `OR` is given in the text of this manuscript, and in the section Discussion and in the section Conclusions, even though it was not presented in the section Results.     

We included in Methods and Results section. Thank you.

Lines 67-68: In the cited reference No. 8 there is no mention of `non-steroidal anti-inflammatory drugs` at all? Explain.   

Thank you for your observation. We tried to emphasize the potential risk that high use of NSAIDS might pose and cited an article that indirectly described UGIB, without actually mentioning it in its text. Thus, we provided another reference, more recent, that better describes the risk. PMID: 33213773

Lines 70-73: The manuscript does not present data on `... death causes in patients admitted for UGIB who also had a COVID-19 infection.`, therefore that aim was not achieved.   

We appreciate your observation. We corrected the phrase.

 Lines 84-88: State all codes, according to ICD, which were used to identify the cases. State which version of the ICD was used.    

Thank you for your observation. We added your recommendations.

Line 91: First of all, it should be explained in detail in the section Materials and Methods whether the patients provided WRITTEN VOLUNTARY consent to participate in the study, or whether they only gave an `Informed consent` for the endoscopic procedures. 

Thank you for your comment. As stated in the manuscript we only obtained written consent for endoscopic procedures. As our study is a retrospective study, written voluntary consent could not be obtained at the time of the examination. In this case we did not perform any changes in the manuscript.

   Line 98: Add to the description of the characteristics of patients also the values of laboratory parameters (such as mean Hb level, etc), which are of importance to the study or are mentioned multiple times in the Discussion in this paper and are not presented in the Results.   

   Thank you for your observation. We added them in Table 3.

Line 135: In the entire section Results, below every table and figure the applied statistical test should be stated.    

We included them. Thank you.

Lines 138-163: The stated data are directly associated to the aims of this manuscript, therefore they had to be systematized and presented in detail on an appropriate Table.  

We agree with your observation and you inserted the table. 

 Line 164-166: Provide data for the vaccinal status of the stated 22 patients.  

Thank you for your observation. We mentioned that only 1 person was vaccinated.  

 Lines 174-178: On Table 2, alike the Table 1, make sure to mention the NUMBER OF PATIENTS for every score, i.e. `mean (SD, No)`.           

We modified the results in the same way as were presented in Table 1.

Lines 174-178: Since `... to assess ... death causes` was stated as an aim of the paper (on Line 70), the causes of death must be stated in Table 2.     

Thank you for your observation. We corrected both in the aim section as well wherever needed because the cause of death was analyzed only by collected data before death. Collecting this type of data was not possible because for most of the patients that died during that time, according to legislative regulations, a necropsy was not mandatory for COVID-infected patients. However, we attached in the supplementary material a table with the cause of death for all 23 patients as mentioned in the patients files.

Lines 174-178: Results presented in Table 2 are not described appropriately in the corresponding text.        

We modified. Thank you.

Lines 179-188, 198-200, 202-204: The causes of death are not stated.  

Thank you for your observation. We corrected both in the aim section as well wherever needed because the cause of death was analyzed only by collected data before death. Collecting this type of data was not possible because for most of the patients that died during that time, according to legislative regulations, a necropsy was not mandatory for COVID-infected patients.

Lines 189-197: In the subsection Statistical analysis not one test for parallelism was stated. State the results of the test of parallelism. 

Thank you for your observation. We agree with your comment and we modified the text in results section as we did not perform any parallelism analysis. We only considered the  

Thank you for your comment. However, we did not take into consideration any parallelism test f, we just emphasized the monthly variation of the difference between cases admitted during the pre-pandemic compared with the pandemic period.

Lines 208-221: In the context of the above mentioned patient’s description, present the characteristics of patients to which this paragraph refers to on a Table.   

Thank you for your observation. The objective of this paragraph was to assess the rate of UGIB in patients who were not admitted with the first diagnosis of UGIB. Similar data was included in previous table 1 and table 4

 Lines 222-308: Reconstruct the entire text of the section Discussion in a way that explanations for differences and similarities in the results of this manuscript and findings of other cited studies are provided.  

Thank you for your comment. We modified the Discussion section and tried to improve the entire section by focusing and commenting on our results in comparison with the available studies published so far.

 Lines 230-231: If the data presented in this sentence represent a part of this manuscript, the statistical methodology described in the subsection Statistical analysis does not allow the use of the term `associated`. Explain.     

We moved the results into the Results section. Thank you.

Lines 231-232: In the section Results in this manuscript it is stated that 23 patients died (see Table 2). In this sentence `Only 2 of the 17 death cases ...` are described. Explain.

The sentence was modified. Thank you.      

Lines 244-247: State in which study were the values recorded as mentioned in this sentence.     

Thank you for your observation. We added the reference there.

Lines 276: In the Discussion, explain why in this study the COVID-19 patients with a significantly lower mean score for Charlson comorbidity index died, and those patients with a significantly higher mean score for Charlson comorbidity index survived COVID-19.  

It was our mistake and we correct it in Table 2 so the COVID-19 patients with a significantly lower mean score for Charlson comorbidity index survived, and those patients with a significantly higher mean score for Charlson comorbidity index died.

Lines 312-314: The stated data are not presented in this manuscript. Explain.        

We present them in the Results section. Thank you.

Again, we appreciate your help and your constructive comments which clearly helped us to improve our paper.

Reviewer 3 Report

The researchers conducted a prospective study of patients admitted between March 2020 and December 2021 because of UGIB and confirmed with COVID-19 infection, compared with those negative for COVID-19.

The main Findings were: A higher mortality rate (55.2%) and a high risk of death (OR 9.04, p <0.0001), were noted in the COVID-19 pandemic. Mortality was almost related to respiratory failure. Admissions for UGIB have decreased by 23.7% during the pandemic.

The Strengths are: prospective design, compared data to patients before and during the epidemic, and compared positive vs. negative COVID-19. In addition, comprehensive data in terms of hospitalization and endoscopy data (severity, causes of bleeding, time of endoscopy …) were collected.

The limitations of the study should include: a small number of COVID-19-positive patients and the study was conducted in a single center.

The conclusion is too long and repeats the part of the results, particularly several numbers of the results were mentioned again.

The above-mentioned limitation should be added, and the conclusion should be shortened and improved.

Author Response

Dear reviewer, we appreciate your comments and observations and you modified the manuscript accordingly.

The researchers conducted a prospective study of patients admitted between March 2020 and December 2021 because of UGIB and confirmed with COVID-19 infection, compared with those negative for COVID-19.

The main Findings were: A higher mortality rate (55.2%) and a high risk of death (OR 9.04, p <0.0001), were noted in the COVID-19 pandemic. Mortality was almost related to respiratory failure. Admissions for UGIB have decreased by 23.7% during the pandemic.

The Strengths are: prospective design, compared data to patients before and during the epidemic, and compared positive vs. negative COVID-19. In addition, comprehensive data in terms of hospitalization and endoscopy data (severity, causes of bleeding, time of endoscopy …) were collected.

Thank you for your comment. We modified the manuscript and mentioned the fact that it was a retrospective study.

The limitations of the study should include: a small number of COVID-19-positive patients and the study was conducted in a single center.

We appreciate your comments and mentioned it in the limitations section.

The conclusion is too long and repeats the part of the results, particularly several numbers of the results were mentioned again.

We appreciate your comment. We modified both the discussion and the limitation section.

The above-mentioned limitation should be added, and the conclusion should be shortened and improved.

Round 2

Reviewer 1 Report

The manuscript was changed based on the reviewers' comments, addressing all the recommendations. Still, minor editing changes may be made: abbreviated words in tables must be explained in the legend at the table's base. Also, probably the COVID-positive or negative may be changed in SARS-CoV-2 positive or negative or with COVID-19 and without COVID-19. In Table 3, as all were with COVID-19, the authors may use survivors and deceased only.

In Figure 1 b, what are the units for the number of new cases? How is minus 10 at the end?

Author Response

We agree with your observations and performed the required changes. We highlighted in the manuscript the changes with track changes.

We really appreciate helping us improve our manuscript.

Reviewer 2 Report

Thank you for the opportunity to review the manuscript ID: life-2100344 again. Unfortunately, the authors did not adequately address the issues highlighted in my review. From the aim of this manuscript, through the applied methodology (in particular, the applied statistical methodology used to determine OR), through the Results chapter (data for OR are listed only in the text of revised version of this manuscript, although my previous review stated that they should be presented in the appropriate table since it is one of the outcomes highlighted in this manuscript, that the OR is shown in the text of the revised manuscript only for mortality and not for other outcomes described in this manuscript, etc.), to the Discussion section. The biggest shortcoming of this manuscript is the complete absence of a logical flow, coherence and systematical approach in the presentation of data, starting from the set aim and the described outcomes (`Evaluated outcomes were: hospital mortality rate, rate of re-failure of bleeding and hemostasis, need for transfusion and surgery, and duration of hospitalization ... The analyzed risk factors were ...`), including  the results presented in this manuscript.  

Author Response

Dear reviewer,

We really appreciate your comments as they really helped us improve our paper.

As mentioned in the previous comment “Lines 208-221: In the context of the above-mentioned patient’s description, present the characteristics of patients to which this paragraph refers to on a Table.” patients’ characteristics were included according to the comparison made. To be clearer the data for rebleeding, hemostasis, need for transfusion, and surgery was included in other tables (1 and 4). As patient mortality was the only outcome statistically significant and with only half of the patients undergoing endoscopy we hope you understand our reason for not introducing another table. We would have repeated the data.

The chapter “2.4 Outcomes” follow all standard information used for UGIB assessment, however, our main objective was focused on mortality, as we had a low number of patients with UGIB and COVID-19 infection.

We also modified the abstract of the manuscript to better point out our findings.

Round 3

Reviewer 2 Report

Thank you for the opportunity to re-review manuscript ID: life-2100344 again. The authors did not make significant corrections in the version of the paper that was submitted to me for a new re-review regarding all of the issues highlighted in my re-review. But, the authors provided some explanations.

·         Since the paper deals with a very current and important topic, it is necessary for the authors to include in the definitive version of the paper all the explanations they provided in the ``Response Letter'' regarding my remarks, that is, for mortality and for the other outcomes in this manuscript. This is particularly important for explanations for determining ORs for all outcomes.

·         The explanations given by the authors in the `Response Letter' can be partly entered in the section `Methods', and partly in the section `Results'.

·         Consequently, if such notes/explanations were not clearly indicated in the final version of the work, it would remain an open question for the readers whether the goals of the work were achieved (in the title of the paper it is written outcomes instead of mortality, the aims of the paper in the new version of the abstract were not clearly defined). Namely, when the goals are clearly defined, and the authors believe that only results where statistical significance has been achieved should be presented, an appropriate explanation must be written in the text of the paper. Thanks to the authors for the answer.  

Author Response

Dear reviewer,

We included a new table in the manuscript focusing on the primary outcomes between SARS-CoV-2 positive and negative patients with UGIB. We agree with your observation and believe it will aid our manuscript to express our objective and results better.

We really appreciate taking the time to help us improve our work!